# Increasing Selenium and Vitamin E in Dairy Cow Milk Improves the Quality of the Milk as Food for Children

**DOI:** 10.3390/nu11061218

**Published:** 2019-05-29

**Authors:** Arlindo Saran Netto, Márcia Saladini Vieira Salles, Luiz Carlos Roma Júnior, Silvia Maria Franciscato Cozzolino, Maria Teresa Moi Gonçalves, José Esler de Freitas Júnior, Marcus Antonio Zanetti

**Affiliations:** 1Department of Animal Science, Faculty of Animal Science and Food Engineering, University of São Paulo, Pirassununga-SP 13635-900, Brazil; mzanetti@usp.br; 2São Paulo Agribusiness Technology Agency, Ribeirao Preto-SP 14000, Brazil; marcia.saladini@gmail.com (M.S.V.S.); lcroma@iz.sp.gov.br (L.C.R.J.); 3Faculty of Pharmaceutical Sciences FCF, University of São Paulo, São Paulo -SP, 05508-000, Brazil; smfcozzo@usp.br; 4Anhanguera Educacional S/A, Leme-SP 13614-320, Brazil; tereza.goncalves@unianhanguera.edu.br; 5Department of Animal Science, School of Veterinary Medicine and Animal Science, Federal University of Bahia, Salvador-BA 40170-010, Brazil; jose.esler@ufba.br

**Keywords:** cholesterol, metabolism, nutrition, oil

## Abstract

In this study, we investigated the beneficial effects of milk biofortified with antioxidants on the health of children. Two experiments were conducted: experiment one evaluated the milk of 24 Jersey dairy cows (450 ± 25 kg of body weight (BW); 60 ± 30 days in milk dry matter intake (DIM)) given different diet treatments (CON = control diet; COANT = diet with vitamin E and selenium as antioxidants; OIL = diet with sunflower oil; and OANT = diet with sunflower oil containing more vitamin E and selenium as antioxidants), and experiment two evaluated the effect of the milk produced in the first experiment on the health of children (CON = control diet; COANT = diet with vitamin E and selenium as antioxidants; OIL = diet with sunflower oil; OANT = diet with sunflower oil containing more vitamin E and selenium as antioxidants; and SM = skim milk). One hundred children (8 to 10 years old) were evaluated in the second experiment. Blood samples were collected at 0 days of milk intake and 28 and 84 days after the start of milk intake. The cows fed the COANT and OANT diets showed greater selenium and vitamin E concentrations in their milk (*p* = 0.001), and the children who consumed the milk from those cows had higher concentrations of selenium and vitamin E in their blood (*p* = 0.001). The platelet (*p* = 0.001) and lymphocyte (*p* = 0.001) concentrations were increased in the blood of the children that consumed milk from cows fed the OANT diet compared to those in the children that consumed SM (*p* = 0.001). The children who consumed milk from cows fed the OIL diet treatment had increased concentrations of low density lipoprotein (LDL) and total cholesterol in their blood at the end of the supplementation period compared to children who consumed SM. The results of this study demonstrate that the consumption of biofortified milk increases the blood concentrations of selenium and vitamin E in children, which may be beneficial to their health.

## 1. Introduction

Milk and milk products have enormous potential to improve the nutrition and livelihoods of hundreds of millions of poor people around the world [1]. Overall, many poor people are unable to grow to their biological potential due to malnutrition, and milk has an essential role in this process.

The benefits of milk consumption and the nutrients present in milk and dairy products are many, but basically these products represent sources of calcium and vitamin D as well as proteins and other essential nutrients. In addition, dairy products supply phosphorus, potassium, magnesium, and vitamins A, B12, and riboflavin [2]. However, milk has demonstrated its potential to affect health through other substances it contains, such as glutathione, a powerful antioxidant that plays a role in the destruction of free radicals involved in oxidative stress [3].

The inclusion of antioxidant and lipid sources in the diets of dairy cows has been recommended as a method of increasing the production and quality of milk and has beneficial effects on the immune system and human health [4,5]. To combat oxidants produced in cells, animals have a sophisticated mechanism employing antioxidants, and some trace minerals and vitamins participate in this mechanism.

Supplementation with selenium and vitamin E promotes important antioxidant activity that protects the cell membrane against reactive oxygen species (ROS) and lipid hydroperoxides [6] and contributes to growth, reproduction, immunity, hormonal synthesis, and tissue integrity [7]. In addition to these effects, selenium and vitamin E can alter and reduce mastitis by decreasing the amount of ROS released by polymorphonuclear neutrophils associated with the inflammatory reaction. An increase in ROS causes damage to mammary tissue, decreases milk production and, in more severe cases, results in the loss of a mammary quarter [8,9].

Thus, the supplementation of dairy cows with antioxidants, in addition to the advantages for animal health, can provide milk biofortified with greater concentrations of selenium and vitamin E, improving children’s health through the effects of antioxidants. Antioxidants are needed to prevent the formation and oppose the actions of reactive oxygen and nitrogen species, which are generated in vivo and cause damage to DNA, lipids, proteins, and other biomolecules. Endogenous antioxidant defenses (superoxide dismutases, H_2_O_2_-removing enzymes, metal-binding proteins) are inadequate to prevent damage completely, so diet-derived antioxidants are important in maintaining health.

The objective of this study was to evaluate the beneficial effects of antioxidants and unsaturated fat in children who consumed milk from cows fed sunflower oil, selenium, and vitamin E. (REFERENCIA)

## 2. Material and Methods

Experiment 1 was carried out in strict accordance with the National Council for the Control of Animal Experimentation (NCCAE) (Protocol number: 012606/2008). The National Council for the Control of Animal Experimentation, which is made up of multidisciplinary group of professionals, has an internal regiment with guidelines that advocate the analysis process of the studies conducted in the University using animals. The points considered in the guidelines address the researcher’s awareness and sensitivity to the levels of pain, distress, and survival instinct of the animals studied, as well as the physical infrastructure, animal welfare and comfort. This study was conducted in the São Paulo Agency for Agribusiness Technology (*Agência Paulista de Tecnologia dos Agronegócios* (APTA) facilities, located in Ribeirão Preto, SP, Brazil. This location (47°51′W and 21°12′S; altitude 646 m) is considered tropical humid, with an annual precipitation of 1,427 mm, a maximum average temperature of 25 °C, and a minimum average temperature of 19 °C. The animals were allocated to individual stalls with 18 m^2^ dimensions. Experimental diets were provided twice a day (07:00 am and 14:00 pm), targeted to have 10% leftovers. The feed offer was adjusted daily, based on intake from the previous day. All conditions of temperature and animal welfare were respected.

Experiment 2 was carried out in strict accordance with the Committee for Ethics in the Use of Human Beings (approval number: 88/2008) of Anhanguera Educacional, Brazil. After analyzing all the normative documents, the research was analyzed by professionals in order to be approved. The study protocol was recorded in the World Health Organization (WHO) International Clinical Trials Registry Platform (ICTRP) and approved by the Brazilian Registry of Clinical Trials (Protocol number: RBR-2bzdm3).

### 2.1. Study Design

Experiment 1: Twenty-four Jersey dairy cows (450 ± 25 kg of body weight (BW); 60 ± 30 dry matter intake (DIM); mean ± standard deviation) were used. Jersey cows were chosen because of the high amount of solids in the milk of this breed in order to represent a greater alteration of the concentrations of antioxidants in the biofortified milk. The experimental diets were as follows: CON = control diet; COANT = diet with vitamin E and selenium as antioxidants; OIL = diet with sunflower oil; and OANT = diet with sunflower oil containing more vitamin E and selenium as antioxidants. The experimental period consisted of 14 days of adaptation to the diets, followed by 12 weeks of experimentation. The treatments were defined because selenium and vitamin E are important antioxidants whose concentrations in milk are known to be changeable, and the oil was added to verify if it improves the efficiency of vitamin E, given that vitamin E is a liposoluble vitamin. The diet was a mixture of ingredients (concentrate + corn silage), and consumption was monitored daily by weighing the food before delivery and the leftovers the next day.

The whole milk was pasteurized and stored in hermetically sealed plastic containers, previously identified according to the treatments and conditioned at 4 °C. A pool of samples of pasteurized milk from each treatment was collected for the chemical analyses.

Experiment 2: One hundred children between 8 and 10 years old were used to evaluate the biofortified milk obtained in the experiment one. The children received whole milk, with the exception of only one group who received skim milk. The diets of the children were the same, except the type of milk that were our treatments. At school we had a responsible nutritionist and a nutritionist was also put in place specifically to supervise the preparation of all of the children’s meals. The study was randomized. The treatments had the same composition with the exception of the effect of the antioxidants resulting from the diet of the cows. Cows that received antioxidants in the diet with selenium and vitamin E increased the concentration of these antioxidants in milk, according to the results in Table 1. The persons responsible for the children signed a free and informed consent document (FICD), and in the meetings with the parents/guardians to sign the term of participation in the project, the importance of offering the milk at home, at the same times and quantities, was explained. The processed milk was provided to children in a full-time school in the municipality of Casa Branca, São Paulo, Brazil, with follow-up conducted by a nutritionist. Two daily liquid preparations of 250 mL of milk from cows fed different diets were given to the children: (1) milk produced by cows fed the control diet (CON), (2) milk produced by cows fed the sunflower oil diet (OIL), (3) milk produced by cows fed the diet with vitamin E and selenium as antioxidants (COANT), (4) milk produced by cows fed the diet with sunflower oil containing more vitamin E and selenium as antioxidants (OANT), and (5) skim milk (SM).

All of the abovementioned preparations contained the same amount of milk and were designed to contain milk fat as the only source of fat and to contain similar nutrient amounts, except for the group that received skim milk, which had a lower percentage of fat. The preparations were given at breakfast at 7:00 am (250 mL) and at 3:00 p.m. (250 mL) Monday to Friday. On Friday, the children also took home the equivalent of 1 liter of milk per inhabitant of the residence of the same allocated treatment, thus allowing for the intake of 500 mL of milk/day over the weekend.

### 2.2. Data Assessment and Chemical Analysis

Samples of milk (250 mL) were collected on days 0, 28, and 84 of the trial period for the determination of selenium and vitamin E concentrations. After collection, the samples were sent to the Laboratory of Minerals of the Faculty of Animal Science and Food Engineering of the University of São Paulo, Faculty of Veterinary Medicine/USP and stored in a freezer at −20 °C in black plastic bags to protect them from light for subsequent analyses. For the analysis of selenium, wet digestion was performed with a nitric-perchloric mixture of samples, and subsequent reading by fluorometric analysis was performed according to sensitization by diaminonaphthalene [10]. For the determination of the values of vitamin E, milk samples were prepared for analysis by means of high-performance liquid chromatography (HPLC). Vitamin E was determined by liquid chromatography according to the methods of Liu et al. [11]. One gram of whey of milk was weighed into a test tube with 7.3 mL of the saponification solution (11% *v*/*v* KOH, 45% *v*/*v* H 2 O, 55% *v*/*v* ETOH and 0.25 g Vitamin C/sample). The samples were vortexed and the tubes were capped before being heated in a 78 °C water bath for 7 min. The samples were then vortexed again and placed in the water bath for an additional 7 min. For extraction, the samples were vortexed again and cooled in cold water. Four milliliters of isooctane was added to each sample, and after being capped with stoppers, the tubes were vortexed for 2 min to extract vitamin E. The tubes were rested to separate the isooctane from the water, which was transferred to a vial for HPLC and stored at room temperature until the samples were analyzed with HPLC. Calculations were performed using the following formula: LV (ppm) = μg/mL isooctane × 4 mL/g of blood/milk. Blood collections from children used syringes with volumes of 10 mL, held also at 0, 28, and 84 days. Samples were centrifuged at 2000× *g* for 15 min. For the analysis of glucose, total cholesterol, triglycerides, high density lipoprotein (HDL) cholesterol, low density lipoprotein (LDL) cholesterol, urea, and creatinine, Laborlab commercial kits (Laborlab^®^, Guarulhos, São Paulo, Brazil) were used with a Jaffe^®^ automatic analyzer. Erythrograms and hemograms were determined using a Scatter Laser^®^ analyzer.

### 2.3. Statistical Analysis

For the evaluation of the variables related to the animals, the experimental design was randomized, and the results were analyzed using PROC MIXED (SAS Inst., Inc., Cary, NC, USA), with Tukey’s test for significant results in the analysis of variance. In the evaluation of the variables related to children, the following model was considered:
Y*_ijk_* = *μ* + Diet_i_ + Time*_j_* + (Diet × Time)*_ij_* + e*_ijk_*
where Y*_ijk_* = dependent variable; Μ = the general population mean; W*_i_* = effect of collection day, (days) (*i* = 0, 28, and 84 days) with day as a repeated factor; Diet*_j_* = effect of the diet (*j* = 1, 2, 3, and 4); and e*_ijk_* = the unexplained residual element assumed to be independent and normally distributed. The data collected over time were analyzed as measures repeated in time with the following sampling times: 0, 28, and 84 days. The fixed effects of treatments, days of collection, and interaction of days × treatments were considered. The means were obtained through LSMEANS, and the significance level was 5%.

## 3. Results

### 3.1. Selenium and Vitamin E Concentrations in the Blood and Milk of Dairy Cows

Cows fed the COANT and OANT diets had higher concentrations of selenium in their blood and milk (*p* = 0.001) and higher concentrations of vitamin E in their milk (Table 1) than cows fed CON and OIL diets.

### 3.2. Hemoglobin, Selenium, and Vitamin E Concentrations in Children’s Blood

Children who consumed milk produced by cows fed the COANT and OANT diets showed higher concentrations of selenium and vitamin E in their blood (*p* = 0.001, Table 2). The concentrations of selenium and vitamin E increased after 28 days of milk consumption compared to their concentrations on the first day of collection (*p* = 0.001).

The treatments had no effects on the blood concentrations of white cells, red cells, hemoglobin, hematocrit, platelets, segmented neutrophils, eosinophils, lymphocytes, or monocytes on any sampling day (Table 2).

There was no difference in the blood concentration of selenium at 28 days in the children who consumed milk from cows fed the CON, OIL, or SM diets. The children who consumed milk from cows fed the CON and SM diets had similar blood vitamin E concentrations at 28 (*p* = 0.001) days. However, at 84 days, the blood concentrations of vitamin E in children who consumed milk from cows fed the COANT and OANT diets were higher than in those who consumed milk from cows fed the OIL and SM diets (*p* = 0.001).

At 28 and 84 days, there was no difference in the selenium or vitamin E concentrations in the blood of children who consumed milk from cows fed the COANT and OANT diets. When the averages were observed throughout all experimental days, the milk from cows fed different diets had no effects on white cell concentrations (*p* = 0.457), red cells (*p* = 0.052), segmented neutrophils (*p* = 0.119), eosinophils (*p* = 0.050), or monocytes (*p* = 0.670) (Table 3). Children who consumed milk from cows fed the OAT diet showed lower hemoglobin concentrations in their blood during the experimental period than children who consumed skim milk (*p* = 0.001, Table 3). However, there was no difference in hemoglobin concentrations between children who consumed milk from cows fed the CON, OIL, COANT, and SM diets. Children who consumed milk from cows fed the CON, OIL, OANT, and COANT diets had higher platelet concentrations than the children who consumed the SM diet.

Collection days affected white blood cell concentrations (*p* = 0.034). On average, the white cells were reduced after 28 days of supplementation for all treatments. Similarly, collection days affected eosinophil concentrations (*p* = 0.029). After 28 days of supplementation, eosinophil concentrations were increased.

### 3.3. Blood Metabolites

Children who consumed milk from cows fed the diets containing oil had higher total cholesterol concentrations among the evaluated diets (*p* = 0.015, Table 4); however, there were no differences in total cholesterol concentrations among children who consumed milk from cows fed the other diets at 84 days of milk supplementation. The milk from cows fed different diets did not affect glucose, cholesterol, triglycerides, HDL, uric acid, urea, creatinine, or LDL concentrations at day 0 or on the concentrations of glucose, triglycerides, HDL, uric acid, urea, creatinine, or LDL at 28 and 84 days of sampling (Table 4). Children who consumed milk from cows fed the OIL diet had higher concentrations of total cholesterol (*p* = 0.001) and LDL cholesterol (*p* = 0.008) compared to children who consumed the SM diet. When the averages were evaluated throughout all experimental days, the milk from cows fed different diets did not affect the concentrations of glucose (*p* = 0.928), triglycerides (*p* = 0.736), HDL cholesterol (*p* = 0.054), uric acid, urea (*p* = 0.323), or creatinine (*p* = 0.179) (Table 5).

The concentrations of glucose (*p* = 0.001) and HDL cholesterol (*p* = 0.001) were increased in the children’s blood at day 28 of milk supplementation, regardless of the cow diet, and were also higher at 84 days of milk supplementation. Cholesterol concentrations (*p* = 0.001) were higher on days 0 and 84 compared to 28 days of milk supplementation. A similar result was observed for triglyceride concentrations (*p* = 0.037), although there was no difference between days 0 and 28 of milk supplementation. The concentrations of uric acid (*p* = 0.001) and urea (*p* = 0.002) were increased with increasing days of milk supplementation, with the highest concentrations observed at 84 days. However, for urea concentrations, there was no difference in those values between collection days 28 and 84.

The children had lower concentrations of LDL cholesterol at 28 days of milk supplementation compared to days 0 and 84 (*p* = 0.001). Similarly, the highest concentrations of selenium in the blood were observed at 28 days of milk supplementation.

## 4. Discussion

### 4.1. Concentrations of Selenium and Vitamin E in Milk

The hypothesis that cows fed the COANT and OANT diets would have increased concentrations of selenium in their milk and blood was confirmed in this study (Table 1). Mean concentrations of selenium in blood and milk of 0.06 and 0.04 μg/mL, respectively (*p* = 0.01), were observed for cows fed the CON and OIL diets, and mean concentrations of selenium in blood and milk of 0.11 and 0.12 μg/mL, respectively, were observed for cows fed the COANT and OANT diets. The objective of this study was to investigate the increase in selenium concentrations in milk, as well as its potential to promote increased protection of the mammary gland against oxidative stress [12]. Paschoal et al. [13] supplemented lactating cows with 5 mg of sodium selenite and 1000 IU of vitamin E in the form of alpha tocopherol acetate and obtained a serum concentration of selenium of 0.09 μg/mL in cows, also demonstrating that supplementation with selenium was effective at increasing the concentration of selenium in blood and milk.

Regarding the effect of the diets on the animals and the effects of the milk that was supplied to the children, selenium and vitamin E supplementation promotes important antioxidant activity that protects cell membranes from reactive oxygen species and lipid hydroperoxides [6]. In addition, supplementation of Se and vitamin E contributes to growth, reproduction, immunity, hormone synthesis, and tissue integrity and has been related to lower incidences of mastitis and somatic cell count and high milk productivity and quality [7].

According to the National Institute of Health, it is recommended that children between the ages of 8 and 13 to receive 40 μg/day of selenium. Welch et al. [14] recommended 55 μg/day of selenium for adults, and the tolerable level is 400 μg/day, as reported by the Institute of Medicine, 2000 [15]. According our results, the consumption of 250 mL of milk obtained from cows supplemented with selenium would provide 30 μg/day of selenium, while the milk of cows that were not supplemented would provide 10 μg/day of selenium. Knowles et al. [16] reported that milk is an important source of selenium and that the intake of 100 g of milk/day can supply more than 10% of the daily requirements of an adult. The concentration of selenium in bovine milk is easy to manipulate, which may be an important way to increase selenium intake by humans. However, strategic supplementation of herds with selenium needs to consider the source of selenium, the dose, and the production system.

Vitamin E is important for health. The main sources of vitamin E are oils and fats, but it is also found in some vegetables, meat fat, poultry, and fish and in lesser amounts in some cereals and dairy products. Charmley et al. [17] found vitamin E concentrations of 1.75 μg/mL and 0.921 μg/mL in cow’s milk when supplemented with 8000 IU of vitamin E and not supplemented, respectively. A very similar value was found in this project, with 1.50 μg/mL in the milk of the cows that received vitamin E supplementation associated with sunflower oil, which would partially meet the daily requirement of vitamin E for children, namely, between 7–9 mg/day. The results confirm that supplementation of cow diets with selenium and vitamin E may increase the levels of these antioxidants in milk and therefore be beneficial to human health.

### 4.2. Hemogram and Concentrations of Vitamin E and Selenium in Children’s Blood

Our investigation began with blood samples taken at the start of the experiment (day 0) when the children had not yet consumed any milk. Thus, the results of the initial concentrations (day 0) of selenium and vitamin E before beginning milk supplementation show no difference between these components in the blood between the different treatments (selenium, *p* = 0.983, and vitamin E, *p* = 0.159) or in any component of the whole blood count (Table 2) or biochemical metabolites (Table 4).

The significant increase in blood concentrations of vitamin E (*p* = 0.001) and selenium (*p* = 0.001) in children who consumed milk from cows fed the OANT and COANT diets at 28 and 84 days of milk supplementation may be attributed to higher concentrations of these antioxidants in the milk provided (Table 1). The milk produced by the cows fed the COANT and OANT diets presented a 100% increase in selenium concentrations when compared to the milk of the cows fed the CON and OIL diets (Table 1). As a consequence of the consumption of this milk, the children absorbed more vitamin E and selenium and therefore had higher blood concentrations of vitamin E and selenium after 28 days of milk consumption (Table 3). The increase in vitamin E concentrations in the blood occurred due to the linkage of absorption of fat-soluble vitamins to the digestion of fats, depending on micellar solubilization, and facilitation by pancreatic bile and lipase [18,19]. Weiss and Wyatt [20] observed increased α-tocopherol concentrations in milk and higher plasma α-tocopherol concentrations in dairy cows fed different levels of vitamin E and fat sources. Regarding the results of selenium, the increase of the mineral in children’s blood represents one of the most commonly used measures to assess the status of selenium in humans [21]. Selenium concentrations in blood and urine reflect the recent intake of selenium. Selenium content analyses of the hair or nails can also be used to monitor long-term intake over months or years. Quantification of one or more selenoproteins (such as glutathione peroxidase and selenoprotein P) is also used as a functional measure of selenium status [22]. Plasma or serum concentrations of 8 micrograms (mcg)/dL or more in healthy people usually meet the needs for selenoprotein synthesis [21].

Blood levels of vitamin E and selenium were increased in children who consumed milk from cows fed the COANT and OANT diets; the values increased at 28 and 84 days. This increase was associated with a decrease in white blood cell concentrations (*p* = 0.001) and an increase in eosinophils (*p* = 0.001). Selenium has numerous functions in the body as a component of approximately 25 selenoproteins already identified in the human body that are important in the regulation of various physiological functions. Selenium has been shown to be an important factor in the immunomodulatory response, and its supplementation has already demonstrated a protective effect against certain cancers, especially in populations where dietary selenium deficiency is present [23,24]. Changes in lymphocyte concentrations caused by vitamin E concentrations may be explained by the fact that lymphocytes are cells that are an integral part of the immune defense system. Therefore, any pathological processes in the body, or processes otherwise affected by the number of blood cells, indicate a reaction of the immune system. Vitamin E acts as a free radical scavenger [25] and can react with superoxide anions, singlet oxygen, and carboxylic compounds [26,27]. In addition, vitamin E concentrates inside the membranes and acts synergistically with ascorbate, as it can sequester radicals by the donation of a hydrogen ion, with the radical formation of tocoferoxil, which is regenerated to its reduced form by vitamin C [28]. Platelets are blood cells formed in the bone marrow from megakaryocytes that fragment, so they are anucleate, that is, devoid of a nucleus. Increased eosinophil concentrations (*p* = 0.001) at 28 and 84 days of milk supplementation may be associated with an increase in activity of the children’s immune system [24].

### 4.3. Biochemical Metabolites in Children’s Blood

Common metabolic parameters can be used to assess homeostasis in animals, as well as its alteration, which could suggest oxidative stress. Some examples are the levels of glucose, non-esterified fatty acids, triglycerides, total proteins, albumin, urea, creatinine, and total cholesterol [29,30].

We did not observe differences in blood metabolite concentrations in the first sampling (day 0) or at 28 days of milk supply. However, at 84 days, children who consumed milk from cows fed the OIL diet showed higher total cholesterol concentrations (*p* = 0.015). Although there was no difference between children who consumed milk from cows fed the CON diet and children who consumed skim milk, the children who consumed skim milk did not present serum cholesterol levels that differed from those observed after 84 days of milk supplementation from cows fed the COANT and OANT diets (Table 3). Some critics caution against the widespread risks of adopting low-fat diets for children [31]. The intake of fat-soluble vitamins allied to a specific type of fat can promote an increased immune response. The glucose results were within the appropriate reference values (70–110 mg/dL), as were results for total cholesterol (<200 mg/dL), triglycerides (<150 mg/dL), HDL (>40 mg/dL), and LDL (<160 mg/dL).

The blood of the children who consumed skim milk showed the lowest values of cholesterol and LDL fraction (*p* = 0.001; Table 5). Studies by the American Academy of Pediatrics (AAP) in 2004 showed that early coronary arteriosclerosis is related to high serum cholesterol, low density lipoprotein (LDL) cholesterol, and very low density lipoprotein cholesterol (VLDL) and low serum concentrations of high density lipoprotein (HDL). Although no apparent risk for nutrient growth or intake is associated with a diet that meets the requirements, health professionals should evaluate each child individually for total fat intake and investigate excessive consumption of low-fat or fat-free foods. A long-term dietary intervention study demonstrated better lipid concentrations and better eating habits in children with high LDL-cholesterol levels [32,33]. In a 3-year interval, dietary total fat, saturated fat, and cholesterol were lower in the intervention group than in the usual care group. The dietary behavioral intervention promoted adherence to a diet with 28% of energy from total fat, <8% from saturated fat, up to 9% from polyunsaturated fat, and <75 mg/1000 kcal cholesterol per day [33]. As children are constantly growing and changing, the periodic assessments can allow some problems to be detected and treated early [34].

Children are one of the most vulnerable groups in the general population, as they are exposed to high health risks during their growth. Adequate nutrition is one of the factors with the greatest impact on children’s health, and nutritional deficits are responsible, directly or indirectly, for more than 60% of the 10 million deaths of children under five years of age. In this context, it is important to emphasize that educational interventions in basic health care and healthy eating associated with biochemical exams and blood counts would have an important contribution to children’s health [34,35].

## 5. Conclusions

Feeding milk with higher concentrations of selenium and vitamin E to children increases the concentrations of these metabolites in their blood and may have beneficial effects on their health by increasing antioxidant function in their bodies.

## Figures and Tables

**Table 1 nutrients-11-01218-t001:** Selenium and vitamin E concentrations in the serum and milk of dairy cows according to diet.

Item	Treatments				SEM ^a^	*p*-Value
	CON	OIL	COANT	OANT		
Se serum, (µg/mL)	0.06 ^b^	0.07 ^b^	0.11 ^a^	0.12 ^a^	0.01	<0.001
Se milk, (µg/mL)	0.04 ^b^	0.04 ^b^	0.08 ^a^	0.08 ^a^	0.01	<0.001
Vit E serum (µg/mL)	4.93 ^c^	4.24 ^c^	5.63 ^b^	6.58 ^a^	0.01	<0.001
Vit E milk (µg/mL)	0.45 ^b^	0.48 ^b^	0.42 ^b^	0.81 ^a^	0.12	<0.001

CON = Control; OIL= Oil; CANT = Oil + Antioxidant; CO+ANT = Control + Antioxidant; O+ANT = Oil + Antioxidant; SM = Skim. ^a^ SEM = standard error of the mean. Values in the same row with a different superscript (a–c) differ significantly.

**Table 2 nutrients-11-01218-t002:** Blood metabolites in children according to treatments and collection days.

Item	Treatments					SEM ^a^	*p*-Value
	CON	OIL	COANT	OANT	SM		
Day 0							
White cells, (V.A./mm^3^)	7600.00	8766.67	5875.00	6757.14	6757.14	351.65	0.135
Red cells, (V.A./mm^3^)	4.65	4.91	4.90	4.94	4.82	0.07	0.727
Hemoglobin, (g/dL)	12.88	12.78	13.12	12.42	13.27	0.17	0.580
Hematocrit, (%)	38.95	39.45	40.80	38.72	40.31	0.38	0.447
Plates (V.A./mm^3^)	283.83	365.33	249.00	296.57	270.29	14.75	0.151
Segmented (V.A./mm^3^)	50.83	58.66	51.75	54.85	48.42	1.67	0.327
Eosinophils (V.A./mm^3^)	3.66	2.66	3.25	3.42	3.71	0.30	0.828
Lymphocytes, (V.A./mm^3^)	38.83	32.50	37.00	34.71	41.00	1.60	0.463
Monocytes, (V.A./mm^3^)	6.66	6.16	8.00	7.00	6.85	0.41	0.821
Vit. E, (mg/mL)	5.48	5.52	5.22	5.24	5.28	0.18	0.983
Se serum, (µg/mL)	0.03	0.04	0.04	0.04	0.04	0.01	0.159
Day 28							
White cells, (V.A./mm^3^)	6558.33	6445.45	6422.22	8238.46	6575.00	389.46	0.518
Red cells, (V.A./mm^3^)	4.61	4.87	4.76	4.90	4.65	0.04	0.207
Hemoglobin, (g/dL)	12.93	12.97	13.04	12.81	13.12	0.10	0.896
Hematocrit, (%)	40.20	40.32	40.82	40.17	40.31	0.27	0.966
Plates (V.A./mm^3^)	276333	313364	266556	313308	256063	8939.15	0.106
Segmented (V.A./mm^3^)	43.66	47.81	51.66	52.53	45.37	1.33	0.154
Eosinophils (V.A./mm^3^)	4.91	4.27	4.33	6.76	4.75	0.39	0.246
Lymphocytes, (V.A./mm^3^)	43.58	40.45	37.00	33.76	41.50	1.19	0.061
Monocytes, (V.A./mm^3^)	7.83	7.45	7.00	6.92	8.37	0.31	0.539
Vit. E, (µg/mL)	5.96 ^c^	7.96 ^b^	8.75 ^a^	10.20 ^a^	5.36 ^c^	0.37	<0.001
Se serum, (µg/mL)	0.04 ^c^	0.04 ^c^	0.11 ^a^	0.07 ^b^	0.03 ^c^	0.01	<0.001
Day 84							
White cells, (V.A./mm^3^)	6750.00	6418.18	6177.78	6884.62	6618.75	233.66	0.917
Red cells, (V.A./mm^3^)	4.72	4.85	4.76	4.85	4.71	0.04	0.686
Haemoglobin, (g/dL)	12.93	12.82	12.96	12.61	13.15	0.10	0.549
Haematocrit, (%)	40.38	39.81	40.34	39.60	40.48	0.25	0.741
Plates (V.A./mm^3^)	295333	311636	286333	328077	264125	9672.36	0.214
Segmented (V.A./mm^3^)	48.75	46.45	47.11	51.46	47.62	1.16	0.697
Eosinophils (V.A./mm^3^)	4.58	3.90	4.22	6.69	6.43	0.52	0.295
Lymphocytes, (V.A./mm^3^)	37.16	41.81	40.77	33.76	38.81	1.02	0.110
Monocytes, (V.A./mm^3^)	9.50	7.90	7.88	8.07	7.12	0.42	0.481
Vit. E, (µg/mL)	5.94 ^c^	7.92 ^b^	8.60 ^a^	10.08 ^a^	5.06 ^c^	0.38	<0.001
Se serum, (µg/mL)	0.04 ^bc^	0.04 ^bc^	0.07 ^a^	0.06 ^a^	0.04 ^c^	0.01	<0.001

CON = Control; OIL= Oil; CANT = Oil + Antioxidant; CO+ANT = Control + Antioxidant; O+ANT = Oil + Antioxidant; SM = Skim. ^a^ SEM = standard error of the mean. Values in the same row with a different superscript (a–c) differ significantly.

**Table 3 nutrients-11-01218-t003:** Blood metabolites in children according to treatments and collection days.

Item	Treatments					SEM ^1^	Days			SEM ^1^	*p*-Value		
	CON	OIL	COANT	OANT	SM		0	28	84		Treat	Days	Treat x Days
White cells, (V.A./mm^3^)	6933	6930	6325	6386	6659	296.4	7130 ^a^	6408 ^b^	6402 ^b^	363.8	0.457	0.034	0.927
Red cells, (V.A./mm^3^)	4.66	4.88	4.78	4.83	4.73	0.05	4.81	4.76	4.78	0.04	0.052	0.737	0.938
Hemoglobin, (g/dL)	12.9 ^ab^	12.9 ^ab^	13.0 ^ab^	12.7 ^b^	13.2 ^a^	0.84	13.1	12.9	12.8	0.11	0.040	0.432	0.984
Hematocrit, (%)	39.9	40.1	40.6	39.5	40.5	0.34	39.9	40.3	40.1	0.27	0.119	0.611	0.837
Plates (V.A./mm^3^)	2914 ^ab^	3238 ^ab^	2801 ^ab^	3171 ^a^	2648 ^c^	116	3042	2851	2971	983	0.001	0.355	0.982
Segmented (V.A./mm^3^)	47.5	50.0	51.1	52.4	47.0	1.69	52.3	48.2	48.2	1.28	0.083	0.051	0.649
Eosinophils (V.A./mm^3^)	4.19	3.60	3.77	5.53	5.02	0.47	3.10 ^b^	5.00 ^a^	5.16 ^a^	0.37	0.050	<0.001	0.748
Lymphocytes, (V.A./mm^3^)	40.2 ^ab^	38.9 ^ab^	38.0 ^ab^	34.7 ^c^	40.4 ^a^	1.14	37.6	39.2	38.4	1.48	0.029	0.616	0.304
Monocytes, (V.A./mm^3^)	8.08	7.39	7.11	7.33	7.54	0.44	6.86	7.51	8.10	0.33	0.670	0.053	0.665
Vitmian E, (µg/mL)	5.79 ^b^	7.13 ^b^	7.52 ^b^	8.50 ^a^	5.23 ^b^	0.16	5.34 ^b^	7.64 ^a^	7.52 ^a^	0.09	<0.001	<0.001	<0.001
Se serum, (µg/mL)	0.04 ^c^	0.04 ^c^	0.07 ^a^	0.05 ^b^	0.04 ^c^	0.09	0.04 ^c^	0.06 ^a^	0.05 ^b^	0.05	<0.001	<0.001	<0.001

CON = Control; OIL= Oil; CANT = Oil + Antioxidant; CO+ANT = Control + Antioxidant; O+ANT = Oil + Antioxidant; SM = Skim. ^1^ SEM = standard error of the mean. Values in the same row with a different superscript (a–c) differ significantly.

**Table 4 nutrients-11-01218-t004:** Blood metabolites in children according to treatments and collection days.

Item	Treatments					SEM ^a^	*p*-value
	CON	OIL	COANT	OANT	SM		
Day 0							
Glucose, (mg/dL)	74.16	77.66	78.54	77.46	75.68	0.98	0.664
Cholesterol, (mg/dL)	161.2	183.6	167.2	165.3	159.3	3.54	0.294
Triglycerides, (mg/dL)	80.16	92.77	76.63	81.07	75.00	3.46	0.614
HDL, (mg/dL)	45.58	51.44	47.45	51.07	50.93	1.43	0.636
Acid uric, (mg/dL)	2.50	2.20	2.30	2.38	2.52	0.07	0.593
Urea, (mg/dL)	20.41	19.11	17.63	15.84	20.68	0.71	0.136
Creatinine, (mg/dL)	0.58	0.56	0.58	0.62	0.58	0.01	0.596
LDL, (mg/dL)	99.66	113.56	104.55	98.00	93.81	2.80	0.265
Day 28							
Glucose, (mg/dL)	80.58	80.22	82.72	83.07	87.87	1.08	0.121
Cholesterol, (mg/dL)	149.5	159.0	144.4	150.0	140.3	3.20	0.476
Triglycerides, (mg/dL)	79.33	69.55	66.54	74.30	72.43	3.14	0.789
HDL, (mg/dL)	50.41	54.88	51.00	58.00	54.68	1.12	0.183
Acid uric, (mg/dL)	3.20	3.13	2.96	2.83	3.01	0.10	0.841
Urea, (mg/dL)	25.83	24.11	24.45	20.07	22.31	1.06	0.475
Creatinine, (mg/dL)	0.65	0.58	0.55	0.59	0.63	0.01	0.085
LDL, (mg/dL)	86.66	89.77	79.63	78.38	71.87	3.26	0.441
Day 84							
Glucose, (mg/dL)	89.25	85.44	87.00	82.61	81.25	1.27	0.230
Cholesterol, (mg/dL)	162.0 ^b^	193.8 ^a^	164.5 ^b^	172.2 ^b^	151.9 ^b^	3.91	0.015
Triglycerides, (mg/dL)	82.41	83.33	89.00	97.92	80.93	4.63	0.758
HDL, (mg/dL)	55.16	62.22	54.72	59.23	57.12	1.41	0.536
Acid uric, (mg/dL)	3.63	3.25	3.32	4.00	3.64	0.11	0.267
Urea, (mg/dL)	20.41	23.55	19.45	21.92	22.37	0.94	0.734
Creatinine, (mg/dL)	0.65	0.63	0.63	0.68	0.65	0.01	0.740
LDL, (mg/dL)	90.50	115.00	91.90	93.46	88.50	3.20	0.112

CON = Control; OIL= Oil; CANT = Oil + Antioxidant; CO+ANT = Control + Antioxidant; O+ANT = Oil + Antioxidant; SM = Skim. HDL = high density lipoprotein; LDL = low density lipoprotein ^a^ SEM = standard error of the mean. Values in the same row with a different superscript (a–b) differ significantly.

**Table 5 nutrients-11-01218-t005:** Blood metabolites in children according to treatments and collection days.

Item	Treatments					SEM	Days			SEM ^a^	*p*-value		
	CON	OIL	COANT	OANT	SM		0	28	84		Treat	Days	Treat × Days
Glucose, (mg/dL)	81.3	81.1	82.7	81.0	81.6	0.69	76.7 ^b^	82.8 ^a^	85.1 ^a^	0.98	0.928	<0.001	0.059
Cholesterol, (mg/dL)	157.6 ^ab^	178.8 ^a^	158.7 ^ab^	162.5 ^ab^	150.2 ^b^	4.07	167.3 ^a^	148.6 ^b^	168.9 ^a^	8.27	<0.001	<0.001	0.902
Triglycerides, (mg/dL)	80.6	81.8	77.3	84.4	76.1	7.79	81.1 ^ab^	72.4 ^b^	86.7 ^a^	10.80	0.736	0.037	0.839
HDL, (mg/dL)	50.3	56.1	51.0	56.1	54.2	2.98	49.2 ^b^	53.7 ^a^	57.6 ^a^	2.66	0.054	<0.001	0.991
Acid uric, (mg/dL)	3.11	2.86	2.86	3.06	3.07	0.32	2.38 ^c^	3.02 ^b^	3.57 ^a^	0.19	0.508	<0.001	0.570
Urea, (mg/dL)	22.2	22.2	20.5	19.2	21.7	2.11	18.7 ^b^	23.3 ^a^	21.5 ^ab^	1.51	0.323	0.002	0.598
Creatinine, (mg/dL)	0.62	0.59	0.59	0.63	0.62	0.03	0.58 ^b^	0.60 ^b^	0.65 ^a^	0.02	0.179	<0.001	0.583
LDL, (mg/dL)	92.2 ^ab^	106.1 ^a^	92.0 ^ab^	89.9 ^ab^	84.7 ^b^	7.54	101.9 ^a^	81.2 ^b^	95.8 ^a^	3.03	0.008	<0.001	0.947

CON = Control; OIL= Oil; CANT = Oil + Antioxidant; CO+ANT = Control + Antioxidant; O+ANT = Oil + Antioxidant; SM = Skimmed. ^a^ SEM = Stander error of the mean. Values in the same row with a different superscript (a–b) differ significantly.

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
