# Peer review of "Increasing Selenium and Vitamin E in Dairy Cow Milk Improves the Quality of the Milk as Food for Children"

_nutrients, 2019, doi:10.3390/nu11061218_

Round 1
Reviewer 1 Report
Title: Increasing selenium and vitamin E in dairy cow milk improves the health of children
In the present study authors have evaluated the effect of different diet treatment enhanced with antioxidant ingredients on cow’s milk. More importantly, authors on a second bases studied the effect of the produced milk on children consumers. They concluded that children which consumed milk with enhanced antioxidant characteristics and were blood tested showed increased blood concentrations of these metabolites (selenium and vitamin E). As a result, it is highlighted from the present study that the diet of a cow can have a major impact on the produced milk and the consumers of this milk highlighting the strong bound between nutrition and human health. It is a very interesting study with good aspects providing very important information for the dairy industry and the consumers. Only some grammatical errors need to be checked.
Author Response
Thanks for the remarks. The article was also reviewed by the American Journal Expert. Certificate attached. But if you have other comments we are available to improve.

Reviewer 2 Report
This paper presents an unusual approach to improving diets through biofortification and reporting results from both changing the diets of cows fed specific diets to achieve certain nutrient profiles from their milk and changing the diets of children to include different milks. However, there are several issues with this paper that need to be addressed:
My overall comment is that I have trouble seeing a clear justification for why this study was conducted and what it tells us about biofortification and its potential to positively impact children's health that we do not already know. A great deal of additional information is needed. Was the primary reason for biofortification to protect dairy cow health or children's health? Why selenium and vitamin E? Do children need more antioxidants? Are the children in this study deficient in either selenium or vitamin E and have a need for more in their diets? Are there potential detriments to biofortification? What population is this biofortification intended for- all children globally? How were 28 and 84 days selected as timepoints for check-ins?
Title: "improves the health of children" is not adequately supported by the content of the paper. Needs to be softened to reflect actual results of this study
Abstract: does not adequately address why this study was conducted or the implications of the results
Introduction: too vague and general- this section starts off talking about the benefits of milk when that is not the focus of the study, biofortified milk is. The data on the nutrient profile of milk is based on U.S. data- milk is fortified with vitamin D in the U.S.- is it elsewhere?
Line 64: how does antioxidant consumption improve children's health? needs a reference
Lines 69-73: more detail than needed; remove
Lines 76-79: how are geographic and climate details relevant? if so, please explain why.
Line 80: spellcheck- "orts"? what does "offer was adjusted daily, based on intake from the previous day" mean?
Lines 83-90: more detail than needed; remove
Line 94: why Jersey cows?
Lines 94-100: how and why were these treatments selected?
Line 104: why was this experiment done in children?
Line 106-7/109-10: redundant with consent form info
Line 111-115: whole milk? what kind of milk was the non-skim milk treatment? what about the rest of the children's diets? was this study double-blind? randomized? what was the nutrient breakdown of each treatment? were they significantly different in any respects? how was compliance evaluated? did children or parents receive instructions to consume milk for the weekend at the same times as during the week?
Please also add more background on demographics of children involved in this study- are the children healthy? overweight? average BMI? are they regular milk drinkers? percentage of girls v boys? how was sample identified? randomized into different treatment groups?
Lines 128-9: copyedit for English language
Line 133: whey or milk?
Line 149: what is "completely randomized"?
Lines 171-173: did you expect a difference in these measures?
Tables: no label for b, c superscript- please add
Discussion needs to be re-organized. Info about the study conducted in cows and the human intervention study are interwoven in a way that is difficult to follow. What is the primary objective of this study and do the results support it?
Line 244: please add original reference for selenium recommendations. Also, what is the recommendation for children? Why was this study conducted in children?
Line 247: is selenium content of milk static or does it change with cow diet, geography, breed, etc?
Line 279: add transition to movming from discussion of vit E to selenium
Line 285: is a level of 8 mcg/dL appropriate for children?
Lines 304-310: this paragraph neesd to be seaprated into sections, based on topic, and focused on its applicability to he study
Line 334: what kind of dietary intervention study?
Line 336: this is an overstatement- recommend rephrasing
Lines 340-351: this seems like info better suited to an introduction. what value does it add here?
Double-check references: 6 does not seem like it's correctly placed. Source 11 refers to vit E method for beef muscle- why used for dairy?
Author Response
Response to Reviewer 2 Comments
Point 1: This paper presents an unusual approach to improving diets through biofortification and reporting results from both changing the diets of cows fed specific diets to achieve certain nutrient profiles from their milk and changing the diets of children to include different milks. However, there are several issues with this paper that need to be addressed:
My overall comment is that I have trouble seeing a clear justification for why this study was conducted and what it tells us about biofortification and its potential to positively impact children's health that we do not already know. A great deal of additional information is needed. Was the primary reason for biofortification to protect dairy cow health or children's health? Why selenium and vitamin E? Do children need more antioxidants? Are the children in this study deficient in either selenium or vitamin E and have a need for more in their diets? Are there potential detriments to biofortification? What population is this biofortification intended for- all children globally? How were 28 and 84 days selected as time points for check-ins?
Response 1: Please provide your response for Point 1. (in red)
The hypothesis for this project was that the supplementation of cows with antioxidants, may allow the production of functional milk with higher concentration of antioxidants in study, selenium and vitamin E, and that if absorbed by humans have beneficial effects for health. Our research group has previously carried out other research projects and reported the beneficial effects of these antioxidants, mainly for the health of the mammary gland of the cows, reducing somatic cell count in milk and increasing the concentration of selenium and vitamin E in milk. The major difference now is whether increasing the concentration of these antioxidants in milk also increases their uptake by humans, as beneficial effects on health are already known.
We had a scientific discussion with doctors, pharmacists and nutritionists and decided to do it with children for the following reasons:
1- Healthy children do not receive medication and would be a good model to verify the absorption, with less interference;
2- The children stayed in full-time school, receiving the meals of breakfast, lunch, afternoon snack and dinner in the school, accompanied with nutritionist, therefore we had a smaller effect of the diets of the houses.
3 - With the children we were able to work with lower age variability and with a very good sample repeatability
We chose the antioxidants selenium and vitamin E because we know from previous projects that these nutrients are possible to be increased in products of animal origin, with ruminants.
The children were not deficient in selenium and vitamin E, but we know that in Brazil diets are deficient in these nutrients and that increasing the intake there is the possibility of a beneficial effect for health, not needing to buy supplements but to ingest with the natural products of our for example dairy products
We chose 0, 28 and 84 days to know the behavior of some analyzed parameters and to verify possibilities as linear or quadratic effect. We got to talk in the research group about more collections as in time 56 days and 112 days, but as it is an invasive methodology for blood collection, we set the times used.
The inclusion of antioxidant in the diets of dairy cows has been recommended as a method of increasing the production and quality of milk and has beneficial effects on the immune system and human health (Staples et al., 2001; Havartine et al., 2006). To combat oxidants produced in cells, animals have a sophisticated mechanism employing antioxidants, and some trace minerals and vitamins participate in this mechanism. Choi et al (2015) showed that the dairy intake is associated with brain glutathione concentration in humans.
Point 2: Title: "improves the health of children" is not adequately supported by the content of the paper. Needs to be softened to reflect actual results of this stud
Response 2: Thank you. The authors agree with the suggestion and suggest the “Increasing selenium and vitamin E in dairy cow milk improves the quality of the milk as food to children”
Point 3: Abstract: does not adequately address why this study was conducted or the implications of the results.
Response 3: Thank you. The authors altered the abstract especially the conclusion.
Point 4: Introduction: too vague and general- this section starts off talking about the benefits of milk when that is not the focus of the study, biofortified milk is. The data on the nutrient profile of milk is based on U.S. data- milk is fortified with vitamin D in the U.S.- is it elsewhere?
Response 4: Thank you. The authors are according with the observations of review. However, the intention initially was to explain about milk source across effects of the change in the diet of dairy cows with the inclusion of antioxidant and lipid sources and the recommendations of the use this method to increase the production and quality of milk with the beneficial effects on the immune system and human health (ex: Witkowska, et al, (2015); J Food Sci Technol. 2015 Oct; 52(10): 6484–6492; Biofortification of milk and cheese with microelements by dietary feed bio-preparations).
Point 5: Line 64: how does antioxidant consumption improve children's health? needs a reference
Response 5: Antioxidants are needed to prevent the formation and oppose the actions of reactive oxygen and nitrogen species, which are generated in vivo and cause damage to DNA, lipids, proteins, and other biomolecules. Endogenous antioxidant defenses (superoxide dismutases, H2O2-removing enzymes, metal binding proteins) are inadequate to prevent damage completely, so dietderived antioxidants are important in maintaining health.
Reference: Halliwell,B. Antioxidants in human health and disease. Annu. Rev. Nutr. 16:33-50, 1996.
Point 6: Lines 69-73: more detail than needed; remove
Response 6: Thank you. The text was excluded.
Point 7: Lines 76-79: how are geographic and climate details relevant? if so, please explain why.
Response 7: Thank you. The text was excluded.
Point 8: Line 80: spellcheck- "orts"? what does "offer was adjusted daily, based on intake from the previous day" mean?
Response 8: Thank you. The text was corrected. “Cows were fed a total mixed ration twice daily, at 07:00 and 14:00, according to the amount of orts from the previous day to maintain a percentage of refusal between 50 and 100 g/kg of the feed offered”.
Point 9: Lines 83-90: more detail than needed; remove
Response 9: Thank you. The text was excluded.
Point 10: Line 94: why Jersey cows?
Response 10: Thank you. The Jersey cows were chosen because of the high amount of solids in the milk of this breed, to represent a greater alteration of the concentrations of antioxidant in the biofortified milk.
Point 11: Lines 94-100: how and why were these treatments selected?
Response 11: The treatments were selected because this is a line of research that we worked on and we did a pilot project to define
Point 12: Line 104: why was this experiment done in children?
Response 12: Because we wanted to use this project, healthy people, without any medication effect and also to have greater standardization with significant number of repetitions
Point 13: Line 106-7/109-10: redundant with consent form info
Response 13: Thank you. The text was excluded
Point 14: Line 111-115: whole milk? what kind of milk was the non-skim milk treatment? what about the rest of the children's diets? was this study double-blind? randomized? what was the nutrient breakdown of each treatment? were they significantly different in any respects? how was compliance evaluated? did children or parents receive instructions to consume milk for the weekend at the same times as during the week?
Response 14: The children received whole milk, only one group received skim milk. The diet of the children were the same, except the type of milk that were our treatments. At school we had a responsible nutritionist and we also put in a nutritionist daily to accompany all children's meals. The study was randomized. The treatments had the same composition, except the effect of the antioxidant and sunflower oil, on the diet of the cows. The treatments had the same composition except the effect of the antioxidant on the diet of the cows. Cows that received antioxidants in the diet with selenium and vitamin E increased the concentration of these antioxidants in milk, according to the results in table 1. Yes, in the meetings with the parents to sign the term of participation in the project, it was even explained the importance of offering the milk at home, at the same times and quantities.
Point 15: Lines 128-9: copyedit for English language
Response 15: Thank you. The text was altered.
Point 16: Line 133: whey or milk?
Response 16: Thank you. “One gram of whey of milk was weighed into a test tube with 7.3 mL of the saponification solution (11% V/V KOH, 45% V/V H 2 O, 55% V/V ETOH and 0.25 g Vit C/sample).”
Point 17: Line 149: what is "completely randomized"?
Response 17: Thank you. The text was altered.
Point 18: Lines 171-173: did you expect a difference in these measures?
Response 18: Yes, our hypothesis is that it would have a difference mainly in relation to time 0, but we would like to know how the behavior of the parameters evaluated between 28 and 84 days would be. We would like to do another period between 28 and 84 days, but we would have to do one more blood collection of the children
Point 19: Tables: no label for b, c superscript- please add
Response 19: Thank you. The text was included. “Values in the same row with a different superscript (a–b) differ significantly.”
Point 20: Discussion needs to be re-organized. Info about the study conducted in cows and the human intervention study are interwoven in a way that is difficult to follow. What is the primary objective of this study and do the results support it?
Response 20: Thank you. The authors investigated the milk source across effects of the change in the diet of dairy cows with the inclusion of antioxidant and lipid sources and the increase of the production and quality of milk. Thus we wrote the discussion following the two points.
Point 21: Line 244: please add original reference for selenium recommendations. Also, what is the recommendation for children? Why was this study conducted in children?
Response 21: Recommended nutrient intakes of selenium (µg/day) derived from WHO-FAO-IAEA for 26-32 µg/day. The study was conducted with healthy children to have no interference with medication use and also because they stayed in school full time, so we had greater control of meals.
Point 22: Line 247: is selenium content of milk static or does it change with cow diet, geography, breed, etc?
Response 22: Thank you. The selenium content of milk change with the diet and the source selenium organic and inorganic. In addition there were others factors that can change selenium content of milk for example breed and lactation phase.
Point 23: Line 279: add transition to moviming from discussion of vit E to selenium
Response 23: Thank you. The text was altered.
Point 24: Line 285: is a level of 8 mcg/dL appropriate for children?
Response 24: Yes, as cited reference 21. Sunde RA, Selenium. In: Coates PM, Betz JM, Blackman MR, eds (2010) Encyclopedia of Dietary Supplements
Point 25: Lines 304-310: this paragraph neesd to be seaprated into sections, based on topic, and focused on its applicability to he study
Response 25: Thank you. Thank you. The text was altered.
Point 26: Line 334: what kind of dietary intervention study?
Response 26: Sorry, I did not understand the question, line 334 talk about concentration at 28 and 84 days of milk supplementation ... but we understand that milk with antioxidants may have a positive effect on children's immune responses, as cited in reference 24.
Point 27: Line 336: this is an overstatement- recommend rephrasing
Response 27: line 336 are discussions concerning the biochemical parameters in children's blood, we think it is important to leave because we have a table with these results
Point 28: Lines 340-351: this seems like info better suited to an introduction. what value does it add here?
Response 28: Thank you. Thank you. The text was altered.
Point 29: Double-check references: 6 does not seem like it's correctly placed. Source 11 refers to vit E method for beef muscle- why used for dairy?
Response 29: Because it is ruminants, the effect of antioxidant activity is similar in ruminants, both for meat and milk production. Reference 6
Reference 11 - The analysis methodology is the same, for meat or milk, with determination by liquid chromatography
We have the English revision certificate by American Journal Expert, but I can not attach more than 1 document and have already attached the article with the changes
Certificate Verification Key: 14F3-DCC3-5B58-EF24-DB69
Thank you for the comments

Round 2
Reviewer 2 Report
Point 10: Line 94: why Jersey cows?
Response 10: Thank you. The Jersey cows were chosen because of the high amount of solids in the milk of this breed, to represent a greater alteration of the concentrations of antioxidant in the biofortified milk.
Thank you for the response. I recommend adding this info to the paper- it is likely that readers will have the same question.
Point 11: Lines 94-100: how and why were these treatments selected?
Response 11: The treatments were selected because this is a line of research that we worked on and we did a pilot project to define
I understand, but the fact that it’s a line of research and you conducted a pilot project is not a sufficient rationale. Why did you do a pilot project and how were the treatments selected for that? This is worth mentioning in the body of the paper to provide justification for your methods.
Response 14: The children received whole milk, only one group received skim milk. The diet of the children were the same, except the type of milk that were our treatments. At school we had a responsible nutritionist and we also put in a nutritionist daily to accompany all children's meals. The study was randomized. The treatments had the same composition, except the effect of the antioxidant and sunflower oil, on the diet of the cows. The treatments had the same composition except the effect of the antioxidant on the diet of the cows. Cows that received antioxidants in the diet with selenium and vitamin E increased the concentration of these antioxidants in milk, according to the results in table 1. Yes, in the meetings with the parents to sign the term of participation in the project, it was even explained the importance of offering the milk at home, at the same times and quantities.
Please include this information in your paper. I asked these questions because I could not find this info in the body of the paper.
Point 21: Line 244: please add original reference for selenium recommendations. Also, what is the recommendation for children? Why was this study conducted in children?
My original comment has not been addressed. Please check your reference-[15] is not an appropriate source for nutrient recommendations. And if you want to include the Institute of Medicine reference (line 266), it needs a reference as well.
Lines 304-310: this paragraph need to be separated into sections, based on topic, and focused on its applicability to the study
I am still finding this section hard to understand. If platelet augmentation occurs due to conditions like rheumatic fever and ulcerative colitis, etc, then why do you also think that it is a response to selenium and vitamin E concentrations in children’s diets? Is augmentation the wrong word?
Point 26: Line 334: what kind of dietary intervention study?
Response 26: Sorry, I did not understand the question, line 334 talk about concentration at 28 and 84 days of milk supplementation ... but we understand that milk with antioxidants may have a positive effect on children's immune responses, as cited in reference 24.
Line 334 from the original study read: “A long-term dietary intervention study demonstrated better lipid concentrations and better eating habits in children with high LDL-cholesterol levels [32, 33].” So my question is: what is the long-term dietary intervention study you’re referring to? This sentence needs to have more context added, so the reader can understand the link between the study you’re citing (refs 32 and 33) and the current study.
Response 27: line 336 are discussions concerning the biochemical parameters in children's blood, we think it is important to leave because we have a table with these results
Line 336 in the original study reads: “As children are constantly growing and changing, periodic assessments allow any problems to be detected and treated early.”
I flagged this sentence because it’s misleading. I recommend- at least- restating to reflect that periodic assessments can allow some problems to be detected and treated early.
Thank you for your responses.
Author Response
Response to Reviewer 2 Comments
Response author 10: Thank you. The Jersey cows were chosen because of the high amount of solids in the milk of this breed, to represent a greater alteration of the concentrations of antioxidant in the biofortified milk.
Response Reviewer: Thank you for the response. I recommend adding this info to the paper- it is likely that readers will have the same question.
Response author: the text was included.
Point 11: Lines 94-100: how and why were these treatments selected?
Response author 11: The treatments were selected because this is a line of research that we worked on and we did a pilot project to define
Response Reviewer: I understand, but the fact that it’s a line of research and you conducted a pilot project is not a sufficient rationale. Why did you do a pilot project and how were the treatments selected for that? This is worth mentioning in the body of the paper to provide justification for your methods.
Response author: ok, included lines 114 -117 . "The treatments were defined because selenium and vitamin E are important antioxidants that we know that it is possible to change their concentration in the milk and the oil was added to verify if it improves the efficiency with vitamin E for being a liposoluble vitamin"
Response author 14: The children received whole milk, only one group received skim milk. The diet of the children were the same, except the type of milk that were our treatments. At school we had a responsible nutritionist and we also put in a nutritionist daily to accompany all children's meals. The study was randomized. The treatments had the same composition, except the effect of the antioxidant and sunflower oil, on the diet of the cows. The treatments had the same composition except the effect of the antioxidant on the diet of the cows. Cows that received antioxidants in the diet with selenium and vitamin E increased the concentration of these antioxidants in milk, according to the results in table 1. Yes, in the meetings with the parents to sign the term of participation in the project, it was even explained the importance of offering the milk at home, at the same times and quantities.
Response Reviewer: Please include this information in your paper. I asked these questions because I could not find this info in the body of the paper.
Response author: the text was included.
Point 21: Line 244: please add original reference for selenium recommendations. Also, what is the recommendation for children? Why was this study conducted in children?
Response Reviewer: My original comment has not been addressed. Please check your reference-[15] is not an appropriate source for nutrient recommendations. And if you want to include the Institute of Medicine reference (line 266), it needs a reference as well.
Response author: Included reference - Institute of Medicine. Dietary reference intakes for vitamin C, vitamin E, selenium, and carotenoids. Washington (DC): National Academy Press; 2000
Lines 304-310: this paragraph need to be separated into sections, based on topic, and focused on its applicability to the study
Response Reviewer: I am still finding this section hard to understand. If platelet augmentation occurs due to conditions like rheumatic fever and ulcerative colitis, etc, then why do you also think that it is a response to selenium and vitamin E concentrations in children’s diets? Is augmentation the wrong word?
Response author: Thank you. You are correct. This information was deleted because it is not correct
Point 26: Line 334: what kind of dietary intervention study?
Response author 26: Sorry, I did not understand the question, line 334 talk about concentration at 28 and 84 days of milk supplementation ... but we understand that milk with antioxidants may have a positive effect on children's immune responses, as cited in reference 24.
Response Reviewer: Line 334 from the original study read: “A long-term dietary intervention study demonstrated better lipid concentrations and better eating habits in children with high LDL-cholesterol levels [32, 33].” So my question is: what is the long-term dietary intervention study you’re referring to? This sentence needs to have more context added, so the reader can understand the link between the study you’re citing (refs 32 and 33) and the current study.
Response author: Ok thank you for explication. "In 3 y interval , dietary total fat, saturated fat, and cholesterol were lower in the intervention group than in the usual care group . The dietary behavioral intervention promoted adherence to a diet with 28% of energy from total fat,<8% from saturated fat, up to 9% from polyunsaturated fat, and <75 mg/1000 kcal cholesterol per day". Information included in article
Response author 27: line 336 are discussions concerning the biochemical parameters in children's blood, we think it is important to leave because we have a table with these results
Response Reviewer Line 336 in the original study reads: “As children are constantly growing and changing, periodic assessments allow any problems to be detected and treated early.”I flagged this sentence because it’s misleading. I recommend- at least- restating to reflect that periodic assessments can allow some problems to be detected and treated early.
Response author: Thank you, the text was altered
